# *Lactobacillus johnsonii* Improves Intestinal Barrier Function and Reduces Post-Weaning Diarrhea in Piglets: Involvement of the Endocannabinoid System

**DOI:** 10.3390/ani14030493

**Published:** 2024-02-02

**Authors:** Zhangzheng Yin, Kaijun Wang, Yun Liu, Yunxia Li, Fang He, Jie Yin, Wenjie Tang

**Affiliations:** 1College of Animal Science and Technology, Hunan Agricultural University, Changsha 410128, China; yinzhangzheng2021@163.com (Z.Y.); kj-wang@foxmail.com (K.W.); liyunxia201003@outlook.com (Y.L.); yinjie@hunau.edu.cn (J.Y.); 2College of Animal Science and Technology, Southwest University, Chongqing 400715, China; yunliu0816@163.com; 3College of Veterinary Medicine, Southwest University, Chongqing 400715, China; 4Animal Breeding and Genetics Key Laboratory of Sichuan Province, Sichuan Animal Science Academy, Chengdu 610066, China

**Keywords:** *Lactobacillus johnsonii*, endocannabinoid system, intestinal health, weaned piglet

## Abstract

**Simple Summary:**

The intestinal health problems of weaned piglets cause serious economic losses. Historically, antibiotics have been used to prevent and treat intestinal problems in weaned piglets. However, the prohibition of antibiotics makes us need to find a safer and effective solution strategy. A large number of studies have shown that the endocannabinoid system, as a lipid mediator signaling system widely distributed in the gastrointestinal tract, interacts with the intestinal microbiota and maintains intestinal homeostasis. In this study, we found that the endocannabinoid system is closely related to the abundance of *Lactobacillus johnsonii* through a piglet antibiotic model. It was further found that *Lactobacillus johnsonii* could improve intestinal health problems and alleviate piglet diarrhea. At the same time, its ability to regulate the endocannabinoid system was verified, and the correlation analysis found that its benefits are partly achieved through the participation of the endocannabinoid system.

**Abstract:**

Probiotic intervention is a well-established approach for replacing antibiotics in the management of weaning piglet diarrhea, which involves a large number of complex systems interacting with the gut microbiota, including the endocannabinoid system; nevertheless, the specific role of the endocannabinoid system mediated by probiotics in the piglet intestine has rarely been studied. In this study, we used antibiotics (ampicillin) to perturb the intestinal microbiota of piglets. This resulted in that the gene expression of the intestinal endocannabinoid system was reprogrammed and the abundance of probiotic *Lactobacillus johnsonii* in the colon was lowered. Moreover, the abundance of *Lactobacillus johnsonii* was positively correlated with colonic endocannabinoid system components (chiefly diacylglycerol lipase beta) via correlation analysis. Subsequently, we administered another batch of piglets with *Lactobacillus johnsonii*. Interestingly, dietary *Lactobacillus johnsonii* effectively alleviated the diarrhea ratio in weaning piglets, accompanied by improvements in intestinal development and motility. Notably, *Lactobacillus johnsonii* administration enhanced the intestinal barrier function of piglets as evidenced by a higher expression of tight junction protein ZO-1, which might be associated with the increased level in colonic diacylglycerol lipase beta. Taken together, the dietary *Lactobacillus johnsonii*-mediated reprogramming of the endocannabinoid system might function as a promising target for improving the intestinal health of piglets.

## 1. Introduction

Early weaning strategies (referring to a weaning age earlier than 21 days) aimed at shortening the slaughter cycle of pigs and improving the reproductive performance of sows have been widely implemented in pig production [1]. However, early weaning often triggers weaning stress syndrome in piglets, leading to various intestinal health problems such as impaired intestinal development, a damaged intestinal barrier function, altered intestinal motility, and weakened intestinal immunity [2,3], posing significant challenges to the pig industry and causing significant economic losses [4,5]. Although antibiotics play a crucial role in preventing intestinal diseases in early weaned piglets, the spread of antibiotic-resistant pathogens and antibiotic residues in food has become a serious problem, which forces us to ban antibiotics [6,7,8]. Sweden banned the use of antibiotics in animal husbandry as early as 1986, while the European Union completely banned the use of antibiotics in 2006, and China has also banned the use of antibiotics in animal husbandry in 2020 [9]. However, this led to various intestinal problems in early weaned piglets in Sweden, resulting in a 50% increase in the incidence of diarrhea and an 8–15% increase in feeding costs [9]. However, similar phenomena also occurred in China. Consequently, the discovery of antibiotic alternatives for mitigating early-weaned-piglet diarrhea holds paramount importance for both animal husbandry and food safety.

The mammalian intestinal microbiota consists of trillions of microbes that play a vital role in maintaining host health, particularly in providing colonization resistance against gastrointestinal disorders [10,11,12,13]. After birth, the structure and function of the gastrointestinal tract of piglets need to adapt to the transition from maternal nutrition (via placenta) to intestinal nutrition (colostrum/milk) [14]. More importantly, this process will be accompanied by the intestinal transition from sterile to bacterial, that is, the dynamic colonization of the external microbiota (such as microbiota from the living environment and sow vagina, skin, and feces) [14]. However, due to the immature development of newborn piglets in all aspects, their intestinal microbiota is unstable and vulnerable to environmental stressors and various pathogens, resulting in intestinal dysfunction [15,16]. The establishment of gut microbiota is more important for the growth and health of piglets, which affects the development and maturation of the gastrointestinal tract, metabolic homeostasis, and immune defense through host–microbe interactions [17,18]. Therefore, routine early probiotic intervention (e.g., with *Lactobacillus* spp. and *Bifidobacterium* spp.) during the very early days of pig life might improve gut microbial colonization, as well as intestinal health and disease resistance [18].

However, previous studies and our recent studies have shown that the mechanism of probiotic intervention on host physiology is extremely complex, and the gut microbiome enhances intestinal health by playing a direct or indirect regulatory role [10,12,19,20,21]. These regulatory mechanisms involve complex systems to control various aspects of intestinal homeostasis (intestinal barrier function and immune regulation, etc.), which form a stable network of connections with the gut microbiota to regulate intestinal homeostasis [22,23]. Emerging evidence suggests that the endocannabinoid system (ECS) is a key link within the brain–gut–microbe axis and acts as a regulator of intestinal homeostasis [24]. The ECS is a lipid-mediated signaling system widely distributed in the gastrointestinal tract that affects a variety of physiological and pathophysiological processes throughout the gastrointestinal track [25]. The ECS mainly includes the molecular mechanisms of endocannabinoid receptors, endocannabinoids, and synthesis and degradation. Its functions and regulatory mechanisms are complex, and it can regulate the energy metabolism, gastrointestinal function, immune system response, liver metabolism, and other organ systems [26]. Other studies and our work have shown that the lipid metabolism is closely related to gut microbiota [20,27,28]. More importantly, numerous studies utilizing antibiotics and germ-free mouse models have demonstrated direct or indirect interactions between the ECS and microbiota [29,30,31]. For example, the intervention of special microbes (such as *Akkermansia muciniphila*) in the intestine of mice induced with high fat can directly regulate the activity of the intestinal ECS, improve the intestinal mucosal barrier function, and reduce the occurrence of related diseases [32]. This indicates that the ECS is closely related to gut microbiota and regulates intestinal health. However, investigations on the core microbiota regulating the intestinal ECS and the specific role of the ECS (especially the synthetic and degradative enzymes) in the intestine of weaned piglets are still limited.

Therefore, we hypothesized that (1) ampicillin (AMP) treatment leads to changes in gut microbiota and the ECS in the intestine of weaned piglets, and specific probiotics could regulate the ECS; and (2) using probiotics found in (1) can improve intestinal health through the ECS. Therefore, the purpose of this study was to find probiotics that regulate the ECS using a piglet antibiotic model and further verify the probiotic intervention model to clarify the specific role of the ECS in regulating piglet intestinal health through probiotics. This study aimed to provide reference data for targeted ECS treatment strategies and provide the evidence and scheme of probiotics to improve the intestinal health of piglets.

## 2. Materials and Methods

### 2.1. Bacterial Strain and Reagents

The *Lactobacillus johnsonii*-BNCC135265 (*L. johnsonii*) strain used in this study was purchased from Beijing Beina Chuanglian Biotechnology Institute (Beijing, China). Unless otherwise stated, bacterial strains were grown in MRS broth (MRSB, Qingdao Hope Bio-technology) or on MRS agar (MRSA) plates at 37 °C, pH = 6.2. Ampicillin was purchased from Dalian Meilun Biotech (Dalian, China).

### 2.2. Animals and Diets

The animal model and experimental procedures used in this experiment were approved by the Hunan Agricultural University Institutional Animal Care and Use Committee (202005).

There were two batches of piglets ([Yorkshire × Landrace] × Duroc), with 16 in each batch. The two batches of piglets were independent of each other. All the piglets were weaned at 21 days of life. In order to explore the relationship between gut microbes and the ECS, a total of 16 piglets (6.87 ± 0.15 kg) were randomly divided into two groups (n = 8): piglets in the experimental group received 1 g/L ampicillin (2 mL) via oral administration for 3 weeks; control piglets received normal saline (2 mL) (Figure 1A). The experiment lasted for 21 days.

In order to verify the relationship between *L. johnsonii* and the ECS and explore the role of the ECS in intestinal health, another 16 piglets (6.79 ± 0.15 kg) were randomly divided into two groups (n = 8): the piglets in the experimental group received *L. johnsonii* (2 mL, 10^8^ CFU/mL) via oral administration for 1 week, the piglets in the control group received normal saline (2 mL), and no treatment was conducted for the next two weeks.

The preparation standards of the *L. johnsonii* and ampicillin were selected by referring to Jun Hu et al. [4]. Piglets were orally administered with a 10 mL syringe at 3 pm every day. Piglets were selected from a commercial farm and come from different litters. All were female piglets and housed individually in pens. The relative humidity in the pig house was controlled at 50–60%. The temperature of the pig house was gradually reduced from 28 °C to 23 °C. Diet nutrient requirements were configured according to national research council (2012) recommendations and provided in powder form. Manual feeding was conducted at four time points (6 am, 10 am, 2 pm, and 6 pm) every day. All piglets were provided free access to basic diets (satisfying the nutritional requirements) and drinking water and slaughtered at the third week after weaning. Samples of jejunum, colon, and their contents from all piglets were collected and stored at −80 °C for further processing.

### 2.3. Diarrhea Index and Growth Performance

The diarrhea index of all piglets was recorded daily beginning on the first day of the challenge. The diarrhea index was scored as follows: 1, normal feces; 2, moist feces; 3, mild diarrhea; 4, severe diarrhea; and 5, watery diarrhea as previously described in [33]. The initial and final body weights were recorded to calculate the average daily gain: average daily gain = (final measured body weight − initial measured body weight)/measured days. Livers, kidneys, and spleens (the superficial fascia was removed and the surface fluids were blotted dry before weighing) were isolated and weighed to calculate the organ index. The calculation formula was organ index (g/kg) = organ weight (g)/body weight (kg) [34].

### 2.4. Reverse Transcription-PCR

Intestinal samples were frozen in liquid nitrogen and ground, and total RNA was isolated using TRIzol reagent (Invitrogen, USA) and then treated with DNase I (Invitrogen, USA). Reverse transcription was conducted at 37 °C for 15 min at 95 °C for 5 s. The primers used in this study were designed according to the pig sequence (Table 1). β-actin or GAPDH were chosen as the house-keeping gene to normalize target gene levels. The PCR cycling condition was 40 cycles at 94 °C for 40 s, 60 °C for 30 s, and 72 °C for 35 s. PCR cycling and relative expression determination were performed according to previous studies [28]. The relative expression was calculated using the 2^−∆∆CT^ method.

### 2.5. L. johnsonii Absolute Quantification

The intestinal contents were long-term stored at −80 °C until extraction. The genomic DNA was extracted from the digesta samples using the QIAamp DNA Stool Mini Kits (Qiagen GmbH, Hilden, Germany). Absolute quantitative real-time PCR was performed to quantify copy numbers of the genes with *L. johnsonii*; the copy number was calculated according to a specific formula. The copy numbers were converted to log 10 for further statistical analysis. For more detailed experimental procedures, refer to the study conducted by Jiao et al. [35]. The *L. johnsonii* quantification used species-specific primers (GHZ10a-F, 5′-CGCTGTTGAATCACCCGT-3′; GHZ10a-R, 5′-TTCGTGGTTGTGGTGTAACTAAA-3′).

### 2.6. Statistical Analysis

All statistical analyses were performed using a *t*-test (the log-transformed data in this study were all in normal distribution) and Pearson correlation analysis using SPSS 20.0 software (SPSS Inc., Chicago, IL, USA). The SPSS Shapiro–Wilk test was used to test the normality of the log transformation data (Part II experiment). The data were expressed as the means ± standard errors of the means (SEM). *p* < 0.05 was considered significant. All figures were generated using GraphPad Prism software 9.5.0.

## 3. Results

### 3.1. AMP Treatment Reduced Diarrhea in Weaned Piglets

By treating piglets with AMP for three weeks (Figure 1A), we first analyzed the effect of AMP on piglet diarrhea, daily weight gain, and organ index. Here, AMP treatment significantly reduced diarrhea in piglets (Figure 1B). However, there was no significant impact on average daily gains (ADG) (Figure 1C) or organ index (liver, kidney, spleen) in all animals (Figure 1D).

### 3.2. AMP Treatment Downregulated the Expression of DAGLB

To explore whether the expressions of ECS-associated genes in the gut are associated with the alteration of microbial abundances caused by antibiotic exposure, jejunal and colonic samples were further tested for endocannabinoid synthesizing and degrading enzymes for subsequent analyses, including endocannabinoid synthase [diacylglycerol lipase beta (*DAGLB*) and N-acyl phosphatidylethanolamine phospholipase D (*NAPE-PLD*)] and degrading enzyme [monoglyceride lipase (*MAGL*) and fatty-acid amide hydrolase (*FAAH*)]. Interestingly, we found that the mRNA level of *DAGLB* was significantly downregulated in the colon, while the *MAGL* expression was upregulated in the jejunum of AMP-fed piglets (Figure 2), indicating a role of gut microbiota in the ECS.

### 3.3. L. johnsonii Was Correlated with the Expression of DAGLB

To explore whether AMP can lead the alterations in the abundance of intestinal *L. johnsonii*, we conducted an absolute quantification analysis on the intestinal *L. johnsonii* levels. AMP treatment reduced the abundance of *L. johnsonii* in the jejunums and colons of weaned piglets (Figure 3A). Notably, Pearson’s correlation analysis showed that the abundance of *L. johnsonii* was significantly positive-correlated with the mRNA level of colonic *DAGLB* (Figure 3B).

### 3.4. Dietary L. johnsonii Reduced Diarrhea in Weaned Piglets

To further verify whether the short-term effect of *L. johnsonii* could affect diarrhea, daily weight gain, and organ index in weaned piglets. We administered *L. johnsonii* treatment to the piglets only during the first week, while the entire experiment lasted for three weeks (Figure 4A). The results show that dietary probiotic *L. johnsonii* significantly ameliorated piglet diarrhea (Figure 4B); however, there was no significant impact on ADG (Figure 4C) or the organ index (liver, kidney, and spleen) of piglets (Figure 4D).

### 3.5. Dietary L. johnsonii Altered the Expressions of Intestinal Health-Related Genes

Subsequently, we further explored the effects of *L. johnsonii* on the gut health of piglets. Among the gut development-related genes, *L. johnsonii* treatment significantly upregulated the mRNA levels of *IGF-1R*, while decreasing the expression of *Preproglucagon* and *EGF* in the jejunum (Figure 5A). But no changes were observed in the colon (Figure 5A). As for gut motility-related genes, the *L. johnsonii* intervention significantly upregulated the mRNA level of *TPH-1* in the colon and reduced the levels of *GLP-1* and *PYY* in the intestines of piglets (Figure 5B). Among the gut barrier-related genes, the *L. johnsonii* treatment increased the expressions of *ZO-1*, *occludin*, and *caludin-1* in the colon (Figure 5C). Moreover, inflammatory cytokines were tested but no differences were observed (e.g., TNF-α, IL-1β, IL-6, IFN-γ, IL-18, IL-12p40, and IL-10) (Figure 5D).

### 3.6. Dietary L. johnsonii-Mediated ECS Reprogramming Was Related to the Intestinal Barrier Function of Piglets

We found that after the *L. johnsonii* treatment, the mRNA expression level of *DAGLB* was selectively upregulated in the colon (Figure 6A). Our data show that the *L. johnsonii*-mediated ECS reprogramming may only occur in the colon (Figure 3 and Figure 6A). We further conducted Pearson’s correlation analysis between ECS-related genes and intestinal health-related genes in two intestinal segments in order to explore the specific role of the ECS in the jejunum and colon. The results show no associations between the ECS and gut health-related genes in the jejunum (Figure 6B). However, we found that in the colon, the expressions of ECS components (especially *DAGLB*) were positively correlated with the level of *ZO-1* (one of the intestinal barrier function-related genes) (Figure 6C). These results indicate that the colonic ECS reprogrammed by *L. johnsonii* might be responsible for the intestinal health of piglets. Together, these findings suggest that the ECS might be an important hub for *L. johnsonii* improving the intestinal barrier function of piglets.

## 4. Discussion

Piglet diarrhea is a multifactorial intestinal disease [36]. The “multiple-hit” hypothesis has emphasized that the “gut microbe-host axis”, including the gut microbiota and gut barrier integrity, serves as a crucial element in the pathogenesis of piglet diarrhea [37]. Due to the prohibition of antibiotics, it is urgent for us to find more safe and effective treatments to control the progression of piglet diarrhea. Probiotic intervention has always been an effective method for replacing antibiotics to improve intestinal health and relieve diarrhea in piglets, but the mechanism is extremely complex [38]. In this study, we reveal the effectiveness of *L. johnsonii* in preventing piglet diarrhea. In addition, the anti-diarrhea effect of *L. johnsonii* was associated with the increased intestinal barrier function, which might be involved in the ECS.

The gut microbiota is a critical environmental determinant of host physiology [39]. Investigations involving germ-free mice or mice subjected to antibiotic-induced gut microbiota depletion have elucidated the roles of microbiota in shaping gut function and enteric neural control mechanisms [40,41]. Although current research on the regulation of the ECS by gut microbiota is still in its infancy, there are still some intriguing results that support the general hypothesis that ECS regulates intestinal homeostasis by interacting with microbiota. In the study on germ-free mice, it was found that the loss of gut microbiota shapes the expression of ECS receptors, as well as biosynthetic and degradative enzymes within the gut [42]. Among them, cannabinoid receptor 1 (CB1) was markedly elevated in the colon of germ-free mice [42]. There were also alterations in the levels of endocannabinoids and other related lipid mediators to a variable degree along the length of the gut [42]. Comparable changes were also noted in studies using antibiotics to deplete the gut microbiota such that modest changes in the anandamine (AEA) levels were noted along the length of the small intestine with no changes in 2-arachidonoylglycerol (2-AG) [43]. Interestingly, antibiotic treatment results in an increased cannabinoid receptor 2 (CB2) expression [43]. Notably, mice were administered with an antibiotic cocktail (including AMP) via an intraperitoneal route, ruling out a possible direct effect of antibiotic exposure on the ECS [43]. These findings suggest that an inseparable relationship between the ECS and gut microbiota exists. However, little is known about the relationship between piglet gut microbes and ECS biosynthetic and degradative enzymes. Here, we used AMP to intervene the gut microbiota of piglet, and we found that the ECS biosynthetic and degradative enzymes changed significantly. Moreover, in our study, we demonstrated that AMP treatment causing lower *L. johnsonii* also shape the expression of ECS components, chiefly DAGLB. Collectively, these findings suggest that the gut microbiota might directly affect all aspects of the ECS, although the exact molecular mechanisms are still not available.

The administration of probiotics is believed to have multiple beneficial effects on intestinal health, including immunomodulation, pathogen inhibition, improved host nutrient metabolism, and intestinal permeability [44]. *L. johnsonii*, as a probiotic, was previously demonstrated to have beneficial effects such as growth promotion and anti-inflammation in animal models [44]. However, our experimental results show that the dietary *L. johnsonii* (as well as AMP treatment) did not improve ADG or the organ index. A study previously reported that a low concentration of *L. johnsonii* intervention had no significant effect on the growth performance of piglets, which was consistent with our study [45]. With the increase in the *L. johnsonii* concentration, the growth performance of piglets was significantly improved [45]. Therefore, we speculate that the treatment concentration may be an important factor; in addition, environmental factors cannot be ignored. We demonstrated that *L. johnsonii* alleviates piglet diarrhea and confers notable improvements in multiple facets of intestinal health (intestinal development, motility, and barrier function). Interestingly, it has been reported that the therapeutic benefits of probiotics might be, to some extent, mediated through the ECS. For example, *Lactobacillus acidophilus* could increase the level of CB2 receptor in intestinal epithelial cells. When animals with visceral hypersensitivity were treated with this probiotic, it gave rise to a pronounced visceral analgesia to colorectal distention, sensitive to CB2 receptor antagonism [46]. Moreover, *Lactobacillus plantarum* reduces despair behavior and increases hippocampal neurogenesis associated with a chronic stress paradigm by altering endocannabinoid levels in the hippocampus [29]. Here, we demonstrated the benefits of *L. johnsonii* on intestinal health and found that it had beneficial effects on piglet diarrhea. Short-term probiotic intervention is a common means to regulate intestinal homeostasis, which can effectively improve intestinal health over a period of time [47]. Our study also proved this, but the duration of this effect needs further confirmation. In addition, from the perspective of ECS biosynthetic and degradative enzymes, our study further verifies the regulatory effect of *L. johnsonii* on the ECS. Specific changes in the gut microbiota in different models (probiotic treatment, high-fat diet, antibiotic treatment, and germ-free mice) selectively alter CB1, FAAH, and MAGL mRNA expressions in the colon [48]. This is consistent with our conclusion that *L. Johnsonii* regulates the ECS, which is restricted to the colon and has no effect in the small intestine. We hypothesize that this phenomenon may be attributed to the higher bacterial load present in the colon. While these intriguing studies provide partial insights into the connection between gut microbiota and the ECS, the precise underlying mechanism remains to be elucidated. This is a focal point of our forthcoming research endeavors.

The endocannabinoids have been linked with intestinal integrity [48]. In a study involving genetically obese mice (*ob/ob*; a model characterized by an altered composition of the gut microbiota and impaired gut-barrier function), blocking CB1 partially restores the distribution and localization of mouse tight junction proteins ZO-1 and occlusion [48]. In wild-type mice, the use of cannabinoid receptor agonists HU-210 causes an increase in intestinal permeability [48,49]. Similarly, in vitro experiments using Caco-2 cells as a model, the addition of HU-210 reduces transepithelial electrical resistance (TEER) and the expressions of the genes encoding ZO-1 and occluding [48]. Notably, these effects are contingent upon CB1 [48]. Furthermore, the addition of AEA and 2-AG, as well as the inhibition of FAAH and MAGL (which are responsible for their degradation) increase intestinal permeability in a CB1-dependent manner [50]. In addition, when cells were exposed to a tumor crossing factor and IFN-γ, thereby creating a model that mimicked inflammatory conditions, analogous findings were obtained [50]. Collectively, these in vivo and in vitro data consistently indicate that AEA as a “door opener” contributes to the disruption of the intestinal barrier function, while 2-AG as a “door keeper” seems to be mainly related to the beneficial effects of the intestinal barrier function in vivo. Of note, our experimental data (*L. johnsonii* treatment) link *DAGLB* to *ZO-1*, which is responsible for the synthesis of 2-AG. In addition, Ma et al. demonstrated that the gene expression of *ZO-1* was downregulated in piglets after AMP treatment, while we demonstrated that the gene expression of *DAGLB* was downregulated [51]. This is consistent with our experimental results, while higher *DAGLB* mRNA levels reduced the intestinal barrier function, which may be one of the indirect reasons for *L. johnsonii* to reduce diarrhea. Therefore, both AMP and *L. johnsonii* can reduce diarrhea, but their *DAGLB* mRNA levels and barrier expressions are inconsistent. The reason may be that some beneficial ways of improving diarrhea (e.g., reduce the abundance of harmful bacteria and improve the intestinal barrier function) are different between them. However, the role of subsequent metabolites (2-AG) has not been proved. Although these intriguing studies provide partial insights into the connection between the ECS and intestinal health of piglets, the precise underlying mechanism remains to be elucidated.

Although the gut microbiota, ECS, and intestinal barrier function have been shown to play roles in intestinal diseases [26], there is currently a lack of research on the interactions between these three components. Collectively, our results indicate that the ECS, particularly DAGLB, appears to play a crucial role in the anti-diarrhea effect of *L. johnsonii* through the maintenance of gut barrier integrity as evidenced by the elevated expression of *ZO-1*. Thus, the *L. johnsonii*-mediated reprogramming of the ECS could function as a promising target for improving the intestinal health of piglets.

## 5. Conclusions

Our results collectively indicate that the ECS, particularly the expression of *DAGLB*, might play a crucial role in the anti-diarrhea effect of *L. johnsonii* through the maintenance of gut barrier integrity. Furthermore, dietary *L. johnsonii* seems to improve the intestinal barrier function of piglets by activating colonic *DAGLB*, leading to an elevated expression of *ZO-1*. In conclusion, given the frequent associations between ECS imbalance, intestinal microbial dysbiosis, and impaired intestinal barrier function with intestinal diseases like piglet diarrhea, dietary *L. johnsonii* is a potential method for improving the intestinal health of piglets.

## Figures and Tables

**Figure 1 animals-14-00493-f001:**
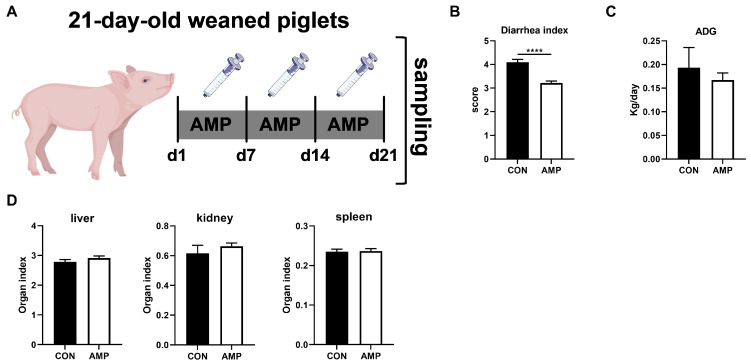
Effects of AMP treatment on the diarrhea, daily weight gain, and organ index of weaned piglets. (**A**) Experiment scheme of piglet antibiotic model; (**B**) Diarrhea index score; (**C**) ADG of piglets; (**D**) Organ index. Values are presented as the mean ± SEM. ****, *p* < 0.0001.

**Figure 2 animals-14-00493-f002:**
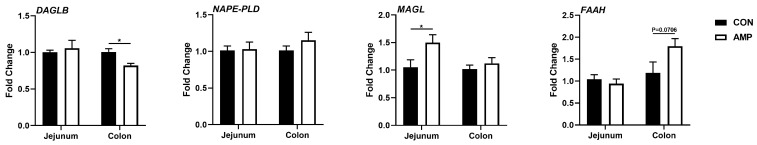
Effect of AMP treatment on the expressions of ECS-related genes. The ECS gene expression profiles. Values are presented as the means ± SEM. *, *p* < 0.05.

**Figure 3 animals-14-00493-f003:**
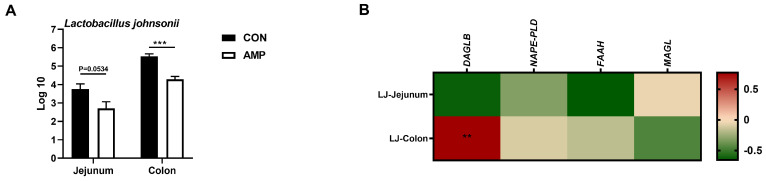
Association analysis between the abundance of *L. johnsonii* and the expression of ECS-related genes (**A**) Absolute quantification of *L. johnsonii* in jejunal and colonic contents; (**B**) Pearson’s correlation between the abundance of *L. johnsonii* and the ECS gene expression. Values are presented as the means ± SEM. Pearson’s correlation coefficient in (**B**) were calculated. **, *p* < 0.01, ***, *p* < 0.001.

**Figure 4 animals-14-00493-f004:**
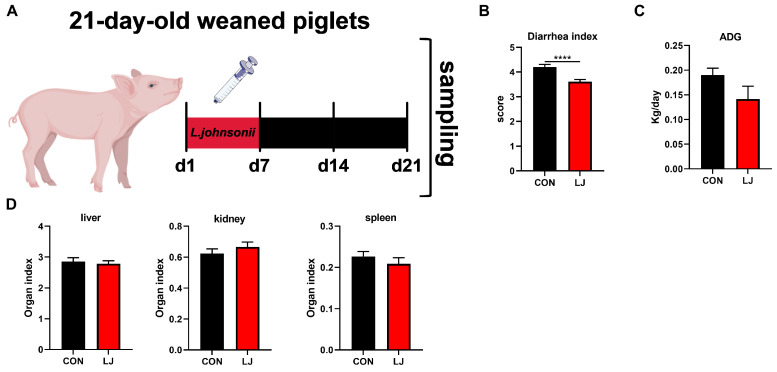
Effects of dietary *L. johnsonii* treatment on the diarrhea, daily weight gain, and organ index of weaned piglets. (**A**) Experiment scheme of *L. johnsonii* intervention; (**B**) Diarrhea index score; (**C**) ADG of piglets; (**D**) Organ index. Values are presented as the means ± SEM. ****, *p* < 0.0001.

**Figure 5 animals-14-00493-f005:**
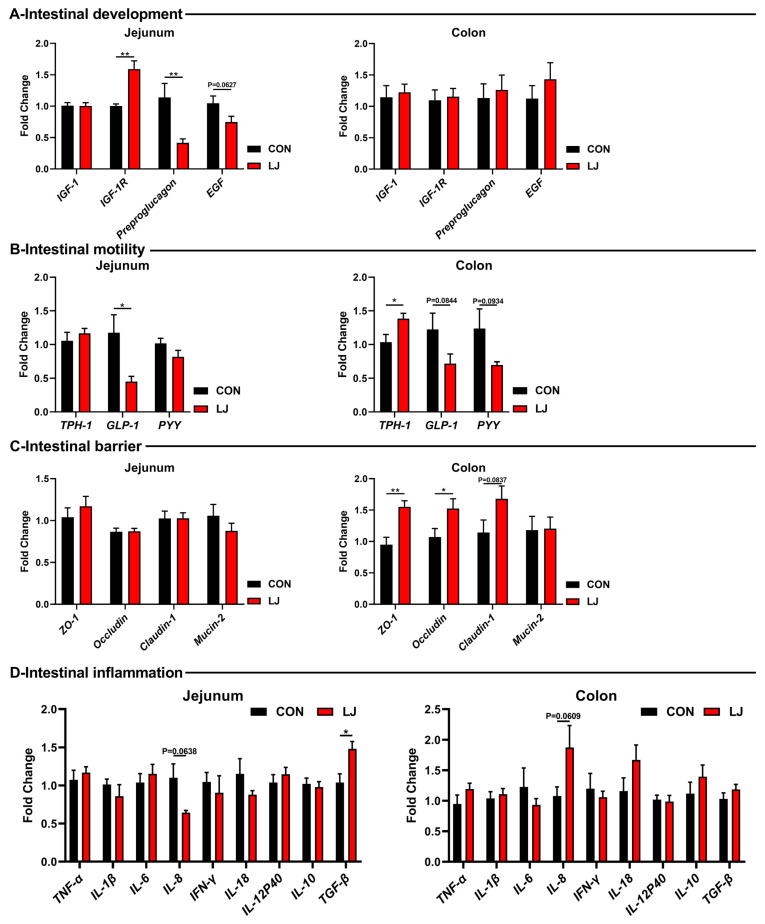
Effect of dietary *L. johnsonii* on the expressions of intestinal health-related genes. (**A**) Intestinal development-related gene expression in jejunum and colon; (**B**) Intestinal motility-related gene expression in jejunum and colon; (**C**) Intestinal barrier-related gene expression in jejunum and colon; (**D**) Intestinal inflammation-related gene expression in jejunum and colon. Values are presented as the means ± SEM. *, *p* < 0.05, **, *p* < 0.01.

**Figure 6 animals-14-00493-f006:**
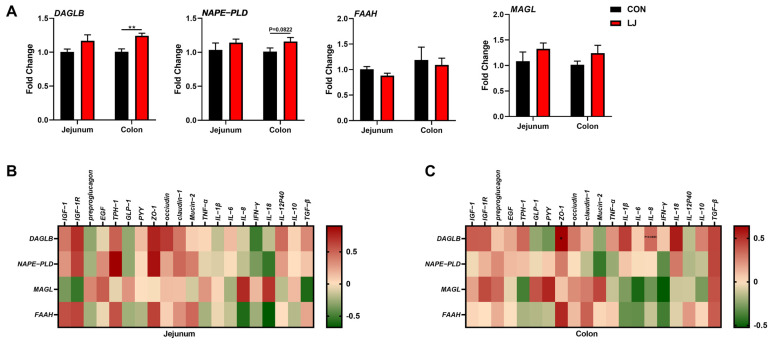
Association analyses between the expression of intestinal health-related genes and the expression of ECS-related genes (**A**) ECS gene expression profile; (**B**) Pearson’s correlation between the ECS gene expression and intestinal health-related gene expression in the jejunum; (**C**) Pearson’s correlation between the ECS gene expression and intestinal health-related gene expression in the colon. Values are presented as the means ± SEM. Pearson’s correlation coefficient in (**B**,**C**) were calculated. *, *p* < 0.05, **, *p* < 0.01.

**Table 1 animals-14-00493-t001:** Primers used for gene expression analysis via real-time PCR.

Gene	Forward Sequence (5′-3′)	Reverse Sequence (5′-3′)
*β-actin*	AGAGCGCAAGTACTCCGTGT	ACATCTGCTGGAAGGTGGAC
*GAPDH*	TCGGAGTGAACGGATTTGGC	TGACAAGCTTCCCGTTCTCC
*FAAH*	TGCCACCGTGCAAGAAAATG	CCACTGCCCTAACAACGACT
*MAGL*	CACTTCTCCGGCATGGTTCT	CGTGAAACGGCGTTGAGC
*NAPE-PLD*	ACCGGCCTCTGAGAAAATGG	AGGGTTAACTGGGGAGACCT
*DAGLB*	TTTGTAATCCCGGACCACGG	GACCTGCCGAGGAATACGGA
*IGF-1*	TCACTGGAGGATGGAATACAGC	CCTGAACTCCCTCTACTTGTGTTC
*IGF-1R*	ATGGAGGAAGTGACAGGGACTA	GTGGTGGTGGAGGTGAAGTG
*Preproglucagon*	ACTCACAGGGCACGTTTACCA	AGGTCCCTTCAGCATGTCTCT
*EGF*	ATCTCAGGAATGGGAGTCAACC	TCACTGGAGGATGGAATACAGC
*TPH-1*	TGGATCTGAACTGGATGCTG	CGGTTCCCCAGGTCTTAATC
*GLP-1*	GACATGCTGAAGGGACCTTTAC	GGCCTTTCACCAGCCAC
*PYY*	AGATATGCTAATACACCGAT	CCAAACCCTTCTCAGATG
*ZO-1*	GAGGATGGTCACACCGTGGT	GGAGGATGCTGTTGTCTCGG
*Occludin*	ATGCTTTCTCAGCCAGCGTA	AAGGTTCCATAGCCTCGGTC
*Claudin-1*	GCATCATTTCCTCCCTGTT	TCTTGGCTTTGGGTGGTT
*Mucin-2*	CTGTGTGGGGCCTGACAA	AGTGCTTGCAGTCGAACTCA
*TNF-α*	CCACGCTCTTCTGCCTACTGC	GCTGTCCCTCGGCTTTGAC
*IL-1β*	AGTGGAGAAGCCGATGAAGA	CATTGCACGTTTCAAGGATG
*IL-6*	CCTCTCCGGACAAAACTGAA	TCTGCCAGTACCTCCTTGCT
*IL-8*	TAGGACCAGAGCCAGGAAGA	AGCAGGAAAACTGCCAAGAA
*IFN-γ*	TCCAGCGCAAAGCCATCAGTG	ATGCTCTCTGGCCTTGGAACATAGT
*IL-18*	TATGCCTGATTCTGACTGTT	ATGAAGACTCAAACTGTATCT
*IL-12P40*	GATGCTGGCCAGTACACC	TCCAGCACGACCTCAATG
*IL-10*	CTGCCTCCCACTTTCTCTTG	TCAAAGGGGCTCCCTAGTTT
*TGF-β*	GAAGATGCTTGGAGCTGAGG	TGGGACTTTGTCTTGGGAAC

*FAAH*, fatty-acid amide hydrolase; *MAGL*, monoglyceride lipase; *NAPE-PLD*, N-acyl phosphatidylethanolamine phospholipase D; *DAGLB*, diacylglycerol lipase beta; *IGF-1*, insulin-like growth factor 1; *IGF-1R*, insulin-like growth factor 1 receptor; *EGF*, epidermal growth factor; *TPH-1*, tryptophan hydroxylase 1; *GLP-1*, glucagon-like peptide-1; *PYY*, peptide YY; *ZO-1*, tight junction protein ZO-1; *TNF-α*, tumor necrosis factor-alpha; *IL-1β*, interleukin-1 beta; *IL-6*, interleukin-6; *IL-8*, interleukin-8; *IFN-γ*, interferon-gamma; *IL-18*, interleukin-18; *IL-12P40*, interleukin-12 subunit p40; *IL-10*, interleukin-10; *TGF-β*, transforming growth factor-beta.

## Data Availability

The data presented in this study are available upon request from the corresponding author. The availability of the data is restricted to investigators based in academic institutions.

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
