# Peer review of "Lactobacillus johnsonii Improves Intestinal Barrier Function and Reduces Post-Weaning Diarrhea in Piglets: Involvement of the Endocannabinoid System"

_animals, 2024, doi:10.3390/ani14030493_

Round 1
Reviewer 1 Report (New Reviewer)
Comments and Suggestions for Authors
Comments to the Authors of manuscript number: animals-2770754 entitled “Lactobacillus johnsonii improves post-weaning diarrhea and intestinal barrier function of piglets: involvement of endocannabinoid system”.
This study investigates the role of the endocannabinoid system mediated by probiotics in the piglet intestine. Antibiotic treatment reprograms the gene expression of the intestinal endocannabinoid system and reduces the abundance of Lactobacillus johnsonii in the colon. The correlation analysis establishes a positive relationship between the abundance of Lactobacillus johnsonii and colonic endocannabinoid system components. Dietary Lactobacillus johnsonii effectively alleviates diarrhea in weaning piglets, improves intestinal development and motility, and enhances the intestinal barrier function. This is evidenced by increased expression of tight junction protein ZO-1, potentially linked to elevated levels of colonic diacylglycerol lipase beta. Overall, the study suggests that dietary Lactobacillus johnsonii-mediated reprogramming of the endocannabinoid system holds promise for enhancing piglet intestinal health.
The manuscript has been improved significantly compared to the previous version. I have no more comments. It can be accepted in this form.
1. L 41- the age should be given. It is not know what is early
2. ECS is presented very well
3. The hypothesis is presented
4. the aim also is presented very well
5. 2.2 the study design is not clear completely. It is not know if the same 16 pigs participated in the two parts. It should be written whether piglets were from the same litter. How they were kept? In one box? Or each piglet is one separated box. Everything should be described in details.
6. Figures presented the study design should be mentioned in 2.2
7. Considering all results, many details are not described in material and methods. All performed analysis should be described in details in part of 2.2.
Author Response
Ref: Submission ID animals-2770754
Dear Editors and Reviewers,
First, we would like to thank you and the reviewers for the thoughtful comments and/or suggestions that have helped us to improve the manuscript. Below are the responses to the general questions raised by the reviewers and all revisions have been marked in red in the main text. We hope the revised manuscript is now suitable for publication in the Animals.
Thank you very much for your attention and consideration. We deeply appreciate your consideration of our manuscript. If you have any queries, please don’t hesitate to contact me at the address below.
Sincerely yours,
Zhangzheng Yin
yinzhangzheng2021@163.com

Reviewer 2 Report (New Reviewer)
Comments and Suggestions for Authors
Animals and diets
More detailed procedures about how ampicillin or L. johnsonii was fed to piglets, as there were only 2ml per pig per day, when and how this was performed
Diarrhea index
As Diarrhea index was scored 0 to 3, then how Figure 1 B and Figure 1 B have values of 4?
There were much more parameters, such as intestinal health-related genes when L. johnsonii was fed, but not when ampicillin was fed, why?
Conclusion
L. johnsonii has the potential for replacing antibiotics to improve the intes-351 tinal health of piglets, but L. johnsonii increased the expression of DAGLB in colon, while AMP decreased the the expression of DAGLB in colon, the data did not support the conclusion.
Comments on the Quality of English Language
Moderate editing of English language required
Author Response
Ref: Submission ID animals-2770754
Dear Editors and Reviewers,
First, we would like to thank you and the reviewers for the thoughtful comments and/or suggestions that have helped us to improve the manuscript. Below are the responses to the general questions raised by the reviewers and all revisions have been marked in red in the main text. We hope the revised manuscript is now suitable for publication in the Animals.
Thank you very much for your attention and consideration. We deeply appreciate your consideration of our manuscript. If you have any queries, please don’t hesitate to contact me at the address below.
Sincerely yours,
Zhangzheng Yin
yinzhangzheng2021@163.com

Reviewer 3 Report (New Reviewer)
Comments and Suggestions for Authors
Author studied “Lactobacillus johnsonii improves post-weaning diarrhea and intestinal barrier function of piglets: involvement of endocannabinoid system.” The description of Materials and Methods needs to be improved. The discussion on the inconsistent effect between AMP and L.J. are missing.
Line 114: The description on live animal management is not clear. For instance, Which guideline was followed for nutrient requirement of the study? How was ampicillin prepared? Were pigs from the second part of the study also harvested at 3 weeks postweaning?
Please specify the gender of pigs used in the study.
Line 128: how often was diarrhea index collected? Was the score taken from all pigs?
Line 158: data normality should be checked after log transformed.
Line 176: why not exam the endocannabinoids expression directly?
Line 181 Is there any explanation on inconsistency between segments?
Figure 3A: AMP reduced the diarrhea of pigs which coincided with lower L johnsonii and DAGLB expression.
Line 194: Please clarify the effect of L johnsonii on ECS and diarrhea.
Line 208: since L.J. reduced diarrhea, why the ADG was lowered in the L.J. group?
Line 215: Treatments were withdrawn at d 7 while tissues were harvested two weeks later. How can we know if L.J. retains its biological function on the gene expression measured? Was the L.J. quantified in second group?
Line 244: It is unclear whether higher or lower the DAGLB is benefited for diarrhea and ADG of pigs based on these two studies.
Line 265: Was intestinal permeability being measured in this trial?
Line 280: How is CB2 expression in this study?
Line 288: How can one exclude the possibility of AMP effect on ECS?
Line 303: since the L.J. and AMP were not coexisted in each trial, comparison of between two treatments are confounded with group.
Comments on the Quality of English Language
Language review is required, especially in the Material and Methods section.
Author Response
Ref: Submission ID animals-2770754
Dear Editors and Reviewers,
First, we would like to thank you and the reviewers for the thoughtful comments and/or suggestions that have helped us to improve the manuscript. Below are the responses to the general questions raised by the reviewers and all revisions have been marked in red in the main text. We hope the revised manuscript is now suitable for publication in the Animals.
Thank you very much for your attention and consideration. We deeply appreciate your consideration of our manuscript. If you have any queries, please don’t hesitate to contact me at the address below.
Sincerely yours,
Zhangzheng Yin
yinzhangzheng2021@163.com

Reviewer 4 Report (New Reviewer)
Comments and Suggestions for Authors
The manuscript titled “Lactobacillus johnsonii improves post-weaning diarrhea and intestinal barrier function of piglets: involvement of endocannabinoid system” try to clarify the specific role of ECS in regulating piglet intestinal health through probiotics. The article is very interesting and contains new data, but the authors made many mistakes.
Title: improves post-weaning diarrhea? the phrase "improve diarrhea" is unfortunate. It should rather be a “reduction of diarrhea”. It should be improved.
Abstract: The abstract does not describe how the research was conducted.
Key words: low letters should be used.
Introduction: There should be spaces here and everywhere before square brackets.
Materials and Methods
L. 108 – Was one strain used or more? Please pay attention to the number of the noun.
L. 111 – was purchased.
L. 114 – It should be rephrased f.e. There are two batches of piglets ([Yorkshire × Landrace] × Duroc), with 16 in each batch. All the piglets were weaned at 21 days of life (6.58±0.15kg).
L. 115 - piglets – remove
According to what standards were the mixtures prepared?
In what conditions were the pigs kept - on what substrate, microclimate conditions. Was there consent from the Local Ethics Committee for the research?
L. 117-123 – The description of the experiment is unclear - are these two separate tests back to back or parallel tests? Why wasn't a control, saline and antibiotic system prepared in parallel? Why was the antibiotic administered for 3 weeks and the probiotic for 1 week? Does this mean that the first experience lasted 3 weeks and the second one lasted a week?
L. 133-134 - The indexes will largely depend on the way the organs are removed. How was it ensured that they were cut identically?
L.164-168 and L. 202-206– repetition from Materials and Methods
L.168 and L. 189-190, 194-195, 209-210, 223-224 - In part, M&M describes the results and does not compare them with others or discuss them. You should also not cite other works (L. 177).
L. 163 - Effects of AMP treatment on growth performance of weaned piglets – the title of the chapter and Figure 1 is not compatible with the data contained therein – f.e. diarrhoea or organs index
Chapter titles are also conclusions, and these should be created after discussing the results as their summary. Figure titles should name the data discussed, not summarize them.
L. 169 – improved? Reduced
L. 170 - average-daily-gain - average daily gains
L. 186 - Figure 2. AMP treatment alters ECS genes expression profile. The ECS genes expression profile. It should be improved.
L. 197 – is – remove italic.
L. 197-8 - Absolute quantification of L. johnsonii in jejunal and colonic contents… How was L. johnsonii quantified? No information about this in section M&M.
L. 212 - Figure 4. Dietary L. johnsonii improves diarrhea in weaned piglet – the title is not adequate t the content – see as Fig. 1.
Disscusion
No reference to performance results and organ weights
Why didn't the addition of antibiotics and probiotics improve weight gain?
References must be improved – sometimes full names sometimes short names of Journals were used.
Author Response
Ref: Submission ID animals-2770754
Dear Editors and Reviewers,
First, we would like to thank you and the reviewers for the thoughtful comments and/or suggestions that have helped us to improve the manuscript. Below are the responses to the general questions raised by the reviewers and all revisions have been marked in red in the main text. We hope the revised manuscript is now suitable for publication in the Animals.
Thank you very much for your attention and consideration. We deeply appreciate your consideration of our manuscript. If you have any queries, please don’t hesitate to contact me at the address below.
Sincerely yours,
Zhangzheng Yin
yinzhangzheng2021@163.com

Round 2
Reviewer 1 Report (New Reviewer)
Comments and Suggestions for Authors
I have no more comments
Author Response
Ref: Submission ID animals-2770754
Dear Reviewers,
Thank you again for the thoughtful comments and/or suggestions that have helped us to improve the manuscript.
Thank you very much for your attention and consideration. We deeply appreciate your consideration of our manuscript. If you have any queries, please don’t hesitate to contact me at the address below.
Sincerely yours,
Zhangzheng Yin
yinzhangzheng2021@163.com
Reviewer 2 Report (New Reviewer)
Comments and Suggestions for Authors
Accept in present form
Author Response
Ref: Submission ID animals-2770754
Dear Reviewers,
Thank you again for the thoughtful comments and/or suggestions that have helped us to improve the manuscript.
Thank you very much for your attention and consideration. We deeply appreciate your consideration of our manuscript. If you have any queries, please don’t hesitate to contact me at the address below.
Sincerely yours,
Zhangzheng Yin
yinzhangzheng2021@163.com
Reviewer 3 Report (New Reviewer)
Comments and Suggestions for Authors
Line 42 Early weaning strategies in swine production refer explicitly to weaning age earlier than 21 days.
Line 125: Was the microbiome studied in this study?
Line 131: The sentence should be in the past tense.
Line 133: Were all pigs kept in the same pen?
Line 136: Diets shouldn’t be in powder form. The powder form is too fine to have a good flow in the feeder.
Line 176: Please specify how the log-transformed data were checked for normality.
Line 208: since L.J. reduced diarrhea, why the ADG was lowered in the L.J. group?
Response: Thanks. In this experiment, ADG decreased but not significantly after LJ treatment. One of the reasons may be the large individual differences. And we added this problem to the discussion. We speculate that the treatment concentration, dose and environment may also be the reasons. Line 300-305
Q: The authors noted factors that could contribute to the study's outcome. Although it was not significantly different, ~ 0.5 kg/d lowered on ADG during the nursery shouldn’t be overlooked. What was the statistics analysis of average body weight?
Line 215: Treatments were withdrawn at d 7 while tissues were harvested two weeks later. How can we know if L.J. retains its biological function on the gene expression measured? Was the L.J. quantified in second group?
Response: Thanks. Thank you for your valuable comments. The detection indicators you mentioned can better support our conclusion. However, in the early stage of the work, we found studies reported in detail the changes in intestinal health of piglets after AMP treatment. We found that some studies reported that both long-term and short-term gavage would increase the abundance of related flora. And it will affect the physiological function of the body. More importantly, we did relevant experiments on mice (unpublished data). We used LJ to treat mice and found that DAGLB was significantly upregulated. In our piglet experiment, we found that intestinal health genes and diarrhea were
improved, and DAGLB was significantly increased. Here are some data and hope to get your approval.
![]() |
Q: The author's previous work does help explain the concern, but why was the AMP only used for 7 days? Moreover, since the data have not been published, they can’t be used as a reference to support the author’s approach. Finding the reference to support the effect of short-term antibiotic exposure on long-term gene expression effect is critical.
Line 288: How can one exclude the possibility of AMP effect on ECS?
Response: Thanks. Our piglet experiment was based on previous mouse and cell experiments (unpublished data). We performed direct AMP treatment in intestinal cells. However, we found that AMP did not directly lead to changes in ECS. Here are some results. We hope to get your approval.
Q: The concern is the same as above. Unpublished data can’t be used as evidence to support the exclusion of AMP on ECS-related gene expressions.
Comments on the Quality of English LanguageThe author indicated that language was reviewed. However, mistakes on grammar still can be found in text.
Author Response
Ref: Submission ID animals-2770754
Dear Editors and Reviewers,
Thank you again for the thoughtful comments and/or suggestions that have helped us to improve the manuscript. Below are the responses to the general questions raised by the reviewers and all revisions have been marked in red in the main text. We hope the revised manuscript is now suitable for publication in the Animals.
Thank you very much for your attention and consideration. We deeply appreciate your consideration of our manuscript. If you have any queries, please don’t hesitate to contact me at the address below.
Sincerely yours,
Zhangzheng Yin
yinzhangzheng2021@163.com

Reviewer 4 Report (New Reviewer)
Comments and Suggestions for Authors
Dear Auror, thank you for correcting the manuscript. I believe that the work can be published in its current version.
Author Response
Ref: Submission ID animals-2770754
Dear Reviewers,
Thank you again for the thoughtful comments and/or suggestions that have helped us to improve the manuscript.
Thank you very much for your attention and consideration. We deeply appreciate your consideration of our manuscript. If you have any queries, please don’t hesitate to contact me at the address below.
Sincerely yours,
Zhangzheng Yin
yinzhangzheng2021@163.com
Round 3
Reviewer 3 Report (New Reviewer)
Comments and Suggestions for Authors
Thank the author's responses.
Author Response
Ref: Submission ID animals-2770754
Dear Reviewer ,
Thank you again for the thoughtful comments and/or suggestions that have helped us to improve the manuscript.
Thank you very much for your attention and consideration. We deeply appreciate your consideration of our manuscript. If you have any queries, please don’t hesitate to contact me at the address below.
Sincerely yours,
Zhangzheng Yin
yinzhangzheng2021@163.com
This manuscript is a resubmission of an earlier submission. The following is a list of the peer review reports and author responses from that submission.
Round 1
Reviewer 1 Report
Comments and Suggestions for Authors
The objective of the study was to evaluate if endocannabinoid system is involed in improvement of intestinal barrier function in piglets by probiotic. That manuscript is in scope of journal, however it needs many changes and explanation before acceptation for publication. Below you can find my suggestion for consideration:
1. I suggest to change title to have the effect of antibiotic treatment.
2. The main criticisms concerning organization and presentation of manuscript. Section „Results” contains informations, which should be inserted i „Materials and Methods” section”
3. Simple Summary is extrimely short. It contains only four stataments.
4. Please, correct the hypotesis and aim.
5. Section „Materials and methods” needs correction. It is mentioned that experiment was performer on eight piglets, but it is probably mistake. Many informations are laking, which are necessary for evaluation of manuscript. For example, condition of ampicilin aplication.
6. What is „organ index”?
7. Please correct figure 5, it is difficult to follow
8. Please, explain why only abudance of Lactobacillus was analysed, not other.
Author Response
Dear reviewer,
Thanks for your comments concerning our manuscript entitled “Lactobacillus johnsonii improves intestinal barrier function of piglets: involvement of endocannabinoid system” (animals-2700652). Those comments were valuable and very helpful. We have read the comments and the instructions for authors carefully. Revised portions are marked in red in the revised manuscript.
Comments and Suggestions for Authors
The objective of the study was to evaluate if endocannabinoid system is involed in improvement of intestinal barrier function in piglets by probiotic. That manuscript is in scope of journal, however it needs many changes and explanation before acceptation for publication. Below you can find my suggestion for consideration:
- I suggest to change title to have the effect of antibiotic treatment.
The original title has been changed to ”Replacement of antibiotics by Lactobacillus johnsonii improves intestinal barrier function of piglets: involvement of endocannabinoid system”.
- The main criticisms concerning organization and presentation of manuscript. Section „Results” contains informations, which should be inserted i „Materials and Methods” section”
We have made changes in the original manuscript as suggested by the reviewers and marked them in red.
- Simple Summary is extrimely short. It contains only four stataments.
We have made more complete changes to the simple summary.
- Please, correct the hypothesis and aim.
We have corrected it.
- Section „Materials and methods” needs correction. It is mentioned that experiment was performer on eight piglets, but it is probably mistake. Many informations are laking, which are necessary for evaluation of manuscript. For example, condition of ampicilin aplication.
We have corrected it and changed it to 8 animals in each group.
- What is „organ index”?
Organ index is the ratio of the weight of an organ of an experimental animal to its body weight.The calculation formula was: organ index (g/kg) =organ weight (g) /body weight (kg). We have made more detailed modifications and cited references.
- Please correct figure 5, it is difficult to follow
We have completely revised figure 5 to present the jejunum and colon results separately.
- Please, explain why only abudance of Lactobacillus was analysed, not other.
We found this phenomenon through the previous experiment of AMP treated mice, so we did the same experiment on piglets to verify the reliability of this conclusion. Through mouse 16S sequencing (unpublished data), we found that Lactobacillus johnsonii was the probiotic with the largest change and the most significant strain associated with ECS. Therefore, we directly detected the content of Lactobacillus johnsonii in piglets for verification.
Reviewer 2 Report
Comments and Suggestions for Authors
this paper presents a test of Lactobacillus johnsonii on digestive tract of weaned piglets, especially note relationship of Lactobacillus johnsonii to ECS. But this paper is not well desinged.1. trow trails of AMP and L johnsonii should be carried out simutaneously.2. there should be another control (known probiotics) for comparision.3. only detecting mRNA is not enough to make sure gene expression, protein detection is also needed.4. person's correlation analysis is not enough to draw the conclusion, before more experimental evidences be provided.
Comments on the Quality of English Languageminor proofread is needed to further improve English writting.
Author Response
Dear reviewer,
Thanks for your comments concerning our manuscript entitled “Lactobacillus johnsonii improves intestinal barrier function of piglets: involvement of endocannabinoid system” (animals-2700652). Those comments were valuable and very helpful. We have read the comments and the instructions for authors carefully. Revised portions are marked in red in the revised manuscript.
Comments and Suggestions for Authors
this paper presents a test of Lactobacillus johnsonii on digestive tract of weaned piglets, especially note relationship of Lactobacillus johnsonii to ECS. But this paper is not well desinged.
- trow trails of AMP and L johnsonii should be carried out simutaneously.
We first find the relationship between microbiota and ECS through AMP treatment, and then we can verify the relationship by probiotic intervention. So we can't do AMP and L johnsonii at the same time.
- there should be another control (known probiotics) for comparision.
We found this phenomenon through the previous mouse AMP model, so we want to know whether the result can be repeated in piglets treated with AMP. Furthermore, we directly located the probiotics with the largest change in 16S and the most significant association with ECS in mice, and explored whether the probiotics could replace antibiotics to improve the intestinal health of piglets and verify whether it partially passed ECS.
- only detecting mRNA is not enough to make sure gene expression, protein detection is also needed.
Based on the findings of previous researchers and the results of our experimental measurements are sufficient to illustrate the contents of this study, and we hope that the reviewers will consider our manuscript.
- person's correlation analysis is not enough to draw the conclusion, before more experimental evidences be provided.
Thank you for your comments. Our current results are intended to give us an enlightenment. We will do a series of protein detection and mechanism exploration in mice and in vitro.
Reviewer 3 Report
Comments and Suggestions for Authors
For both practical and scientific reasons, the submitted manuscript is fascinating and important thematically and practically. Clarification and addition of missing information are, in my opinion, necessary to the publication's text. To assist writers in improving their manuscripts, I've included some suggestions and advice below..
· Abstract – In my opinion, this part of the manuscript should be written in simpler and shorter sentences. The summary should not contain abbreviations and does not need to reference the statistical analysis tools used in the study.
· Section „Introduction” – In my opinion, this part of the manuscript requires correction and addition. The text must be more transparent and justify the topic logically. Try to move more clearly from the issues of early weaning and antibiotic use to the role of the gut microbiota and endocannabinoid system in maintaining gut health. It may also be helpful in bridging the gap between the general context and the specific focus of research on Lactobacillus johnsonii and its relationship to the endocannabinoid system in improving intestinal barrier function. It seems to me that the topic of the endocannabinoid system in maintaining gut health requires precise discussion. It would be useful to strengthen the aim of the work by presenting the research hypothesis and describing the technique for testing it.
· Section „3.4. Intervention with L.Johnsonii improves diarrhea” – In my opinion, it would be valuable to comment on the experimental design, particularly the short duration of probiotic administration. To what extent would the observed changes differ if the probiotic were administered for three weeks, the same period as the antibiotic in the experiments described previously?
· Figure 6 (B) and (C) – In my opinion, these connections require detailed discussion and commentary in the text.
· Section „Discussion” – In my opinion, the discussion of the results could benefit from a clearer structure. I propose to divide the content into separate paragraphs discussing various aspects (effect of L. johnsonii on piglet diarrhea, relationship between gut microbiota and ECS, effect of probiotics on ECS modulation, specific effect of L. johnsonii on ECS and the integrity of the intestinal barrier). The discussion can be strengthened by directly relating the research and findings mentioned to the results of the current study. This will reinforce the relevance and importance of these findings in the context of research on the effects of L. johnsonii on piglet intestinal health and the endocannabinoid system.
· Section „Conclusions” – In my opinion, the conclusions presented provide a clear summary of the main findings of the study and their implications. However, they can be strengthened and refined to increase the transparency and relevance of survey results. It is worth pointing out what is new in the knowledge about the probiotics of L. johnsonii. Can these properties of L. johnsonii be transferred to other probiotics and to what extent? What are the limitations of the research described? What recommendations for further research or pig breeding practice can be suggested based on the research described?
· Entire manuscript – In my opinion, the manuscript would benefit from proofreading by a native speaker.
· Entire manuscript – Please correct "L.Johnsonii" to "L. johnsonii”.
Comments on the Quality of English Language
In my opinion, the manuscript would benefit from proofreading by a native speaker.
Author Response
Dear reviewer,
Thanks for your comments concerning our manuscript entitled “Lactobacillus johnsonii improves intestinal barrier function of piglets: involvement of endocannabinoid system” (animals-2700652). Those comments were valuable and very helpful. We have read the comments and the instructions for authors carefully. Revised portions are marked in red in the revised manuscript.
Comments and Suggestions for Authors
For both practical and scientific reasons, the submitted manuscript is fascinating and important thematically and practically. Clarification and addition of missing information are, in my opinion, necessary to the publication's text. To assist writers in improving their manuscripts, I've included some suggestions and advice below.
- Abstract – In my opinion, this part of the manuscript should be written in simpler and shorter sentences. The summary should not contain abbreviations and does not need to reference the statistical analysis tools used in the study.
We have corrected it.
- Section „Introduction” – In my opinion, this part of the manuscript requires correction and addition. The text must be more transparent and justify the topic logically. Try to move more clearly from the issues of early weaning and antibiotic use to the role of the gut microbiota and endocannabinoid system in maintaining gut health. It may also be helpful in bridging the gap between the general context and the specific focus of research on Lactobacillus johnsonii and its relationship to the endocannabinoid system in improving intestinal barrier function. It seems to me that the topic of the endocannabinoid system in maintaining gut health requires precise discussion. It would be useful to strengthen the aim of the work by presenting the research hypothesis and describing the technique for testing it.
We have corrected it.
- Section „3.4. Intervention with L.Johnsonii improves diarrhea” – In my opinion, it would be valuable to comment on the experimental design, particularly the short duration of probiotic administration. To what extent would the observed changes differ if the probiotic were administered for three weeks, the same period as the antibiotic in the experiments described previously?
This question requires a separate experiment to verify. Based on the comparison between the findings of previous researchers and our findings, our experiment is sufficient to illustrate the research purpose of the manuscript, and we hope that the reviewers will consider our manuscript.
- Figure 6 (B) and (C) – In my opinion, these connections require detailed discussion and commentary in the text.
We have made changes and discussed and commented in more detail.
- Section „Discussion” – In my opinion, the discussion of the results could benefit from a clearer structure. I propose to divide the content into separate paragraphs discussing various aspects (effect of L. johnsonii on piglet diarrhea, relationship between gut microbiota and ECS, effect of probiotics on ECS modulation, specific effect of L. johnsonii on ECS and the integrity of the intestinal barrier). The discussion can be strengthened by directly relating the research and findings mentioned to the results of the current study. This will reinforce the relevance and importance of these findings in the context of research on the effects of L. johnsonii on piglet intestinal health and the endocannabinoid system.
We discussed these four aspects(effect of L. johnsonii on piglet diarrhea, relationship between gut microbiota and ECS, effect of probiotics on ECS modulation, specific effect of L. johnsonii on ECS and the integrity of the intestinal barrie) in more detail.
- Section „Conclusions” – In my opinion, the conclusions presented provide a clear summary of the main findings of the study and their implications. However, they can be strengthened and refined to increase the transparency and relevance of survey results. It is worth pointing out what is new in the knowledge about the probiotics of L. johnsonii. Can these properties of L. johnsonii be transferred to other probiotics and to what extent? What are the limitations of the research described? What recommendations for further research or pig breeding practice can be suggested based on the research described?
Since we have not found a specific mechanism in our study, we cannot know the way in which Lactobacillus johnsonii plays its role. In the later stage, we may need to use engineered bacteria to explore whether it can be transferred, and whether other probiotics have such genes and regulatory functions, which is also the problem we are considering now. Our study has linked the relationship between intestinal microbiota and ECS through mice(Unpublished data) and piglets, but the specific internal mechanism has not been found. We are currently committed to using mice and cells to explore the mechanism, and have constructed knockout mice to verify the function of this gene. According to our current results, Lactobacillus johnsonii is a potential antibiotic replacement strain, and ECS can be used as one of the targets for improving intestinal problems.
- Entire manuscript – Please correct "L.Johnsonii" to "L. johnsonii”.
We have corrected it.
Round 2
Reviewer 1 Report
Comments and Suggestions for Authors
Manuscript has been corrected by Authors. Now it is very clear, that manuscript not represent satisfactory scientific quality to be published in the Journal. It is misunderstaning. Two independent experiments do not give to ability to evaluate the adventages of replacement the antibiotic treatment by probiotic treatment on gut health. It should be one experiment, in which antibiotic and probiotic are applied. It can not be corrected. Additionally, the "Results" section has not been corrected sufficiently accoring to comments. In other places the corrections are also not satisfactory,
Author Response
Thank you for your comment. We are sorry that no effective modification has been made before. We have replied and revised the questions you mentioned before and modified the logic of this article in more detail. Please check the attached file of our reply.

Reviewer 2 Report
Comments and Suggestions for Authors
the revised version dose not improve its quality. my concerns are not answered at all.
Author Response

(The authors gave the same response as above.)

Reviewer 3 Report
Comments and Suggestions for Authors
The manuscript has been effectively revised and enhanced by the authors. All my suggestions were well taken into account by the authors.. The new information and corrections make a substantial contribution, rendering the work more comprehensive and understandable. The authors have adeptly addressed previous comments, leading to an overall enhancement of the manuscript.